# Relationships between External, Wearable Sensor-Based, and Internal Parameters: A Systematic Review

**DOI:** 10.3390/s23020827

**Published:** 2023-01-11

**Authors:** Janina Helwig, Janik Diels, Mareike Röll, Hubert Mahler, Albert Gollhofer, Kai Roecker, Steffen Willwacher

**Affiliations:** 1Institute of Sport and Sport Science, Albert-Ludwigs University Freiburg, 79117 Freiburg, Germany; 2Institute for Advanced Biomechanics and Motion Studies, Offenburg University, Max-Planck Straße 1, 77656 Offenburg, Germany; 3Sport-Club Freiburg e.V., Achim-Stocker-Str. 1, 79108 Freiburg, Germany; 4Institute for Applied Health Promotion and Exercise Medicine, Furtwangen University, 78120 Furtwangen, Germany

**Keywords:** player monitoring, external load, internal load, MEMS, wearable sensors

## Abstract

Micro electro-mechanical systems (MEMS) are used to record training and match play of intermittent team sport athletes. Paired with estimates of internal responses or adaptations to exercise, practitioners gain insight into players’ dose–response relationship which facilitates the prescription of the training stimuli to optimize performance, prevent injuries, and to guide rehabilitation processes. A systematic review on the relationship between external, wearable-based, and internal parameters in team sport athletes, compliant with the PRISMA guidelines, was conducted. The literature research was performed from earliest record to 1 September 2020 using the databases PubMed, Web of Science, CINAHL, and SportDISCUS. A total of 66 full-text articles were reviewed encompassing 1541 athletes. About 109 different relationships between variables have been reviewed. The most investigated relationship across sports was found between (session) rating of perceived exertion ((session-)RPE) and PlayerLoad™ (PL) with, predominantly, moderate to strong associations (r = 0.49–0.84). Relationships between internal parameters and highly dynamic, anaerobic movements were heterogenous. Relationships between average heart rate (HR), Edward’s and Banister’s training impulse (TRIMP) seem to be reflected in parameters of overall activity such as PL and TD for running-intensive team sports. PL may further be suitable to estimate the overall subjective perception. To identify high fine-structured loading—relative to a certain type of sport—more specific measures and devices are needed. Individualization of parameters could be helpful to enhance practicality.

## 1. Introduction

Player monitoring in sports aims at optimizing training adaptations to improve performance and reduce injury risk [1]. Adaptations occur based on psycho-physiological responses to exercise. These internal responses are stimulated by the internal load experienced during exercise; they are difficult to measure directly in a non-invasive way and can only be estimated in typical sports settings. Estimates of internal load and an athlete’s response to exercise are commonly provided by markers of cardiovascular, neuromuscular, or metabolic functioning, e.g., measurements of heart rate (HR) or ratings of perceived exertion (RPE) [2]. Adaptations to training and match demands may be estimated by detecting a change in fitness or fatigue state using, e.g., spiroergometry, cardiopulmonary fitness tests, immunological or hormonal blood markers. Adaptations may be positive or negative, or the fitness state may be maintained. Negative adaptations occur during detraining phases (i.e., off-season) and overtraining, whereas positive adaptations occur after optimal loading and adequate recovery periods.

Internal loading due to sports activities primarily results from movement-related force application demands. Forces need to be applied to the environment to cover running distances, perform changes in movement direction or accelerate or decelerate the body, e.g., during acceleration, stopping, or jumping tasks. Applying forces to the environment results in reaction forces acting on athletes’ bodies, determining the external load stimulus applied to the biological system. Physical external loads applied over time result in different types of internal loads (e.g., mechanical or physiological), which determine the body’s adaptations. Knowledge of the internal response and adaptation to a given dose of external load is crucial for optimal, injury-free training progress. The internal load is influenced by individual factors such as age, gender, training experience, health status, and nutrition [3]. The link between individual characteristics, external load, internal load, exercise-induced responses, and performance adaptations is depicted in Figure 1. In the context of this paper, we refer to internal load, exercise-induced response and adaptations, and individual characteristics as internal parameters. In the same figure, the possibilities to assess these categories are displayed, as they are included in this systematic review.

In this framework, it can be distinguished between four different categories of internal parameter assessments: First, the internal load estimates collected during exercise, primarily made up of HR-based indices and RPE or session-RPE. Second, the exercise-induced responses measured post-exercise due to the delayed response of specific systems to activity, such as creatine kinase (CK), an indicator of muscle damage. Third, the body’s adaptations may be assessed over time (usually tested under standardized conditions, e.g., maximal oxygen uptake (VO_2max_) tests using ergometry). Fourth, the assessment of the current health and fitness status, which, among other parameters such as genetics, age, and gender, make up the individual characteristics. Parameters of each category are included in this systematic review if a relationship to an external load parameter, measured during training or match play using a MEMS device, was assessed.

Most sports science research groups term the responses as exercise and the training or match stimuli as internal and external load, workload, or training load, respectively [1,4,5,6,7]. We acknowledged that this terminology might be misleading considering the mechanical concepts where the load is weight or resistance, which is expressed in Newtons (N), as defined by the Système International d’Unites (SI), as various other research groups have indicated [8,9,10,11]. In order to cover the literature comprehensively, the terms external and internal load were included during the search process and are further used throughout this systematic review, but with their meaning as outlined in Figure 1.

Internal parameters, such as biochemical, hormonal and immunological parameters, are often impractical to collect during training sessions and competitions; doing so might be time and cost-intensive [1]. External load variables can be measured more efficiently and time-effectively. Thus, knowing the relationship between external and internal parameters would be practical to learn about potential dose-response relationships [12,13].

External load variables can be tested in laboratory or field settings. While laboratory settings offer access to accurate gold-standard approaches to quantify external load (e.g., through direct measurements of ground reaction forces (GRFs) using force platforms or the inverse dynamics-based calculation of external joint moments), field settings offer greater ecological validity and the potential to reach larger numbers of athletes.

In the field setting, external load variables can be measured using lightweight, body-worn sensors. With the introduction of global navigation satellite systems (GNSS) devices into the player monitoring market, the research around workload quantification and load monitoring has increased exponentially in the last 15–20 years [14,15]. Next to GNSS, wearable sensor-based load monitoring systems may consist of local positioning systems (LPS), offering higher accuracy in the location of, e.g., team sports athletes in the field of play. Another promising combination of sensor technologies are inertial measurement units (IMUs), commonly combining 3D accelerometers, 3D gyroscope, and 3D magnetometers. Combined in one unit, these systems belong to the group of micro-electro-mechanical systems (MEMS). Commercially available physical activity trackers have gained tremendous interest in the recent decade. “Wearable technology was the top worldwide fitness trend in 2016 and 2017” [16]. Besides physical parameters (i.e., step count), wearables aim to estimate the internal load a person is experiencing. Some smartwatches provide an estimate of, e.g., the metabolic work and power [17]. However, the validity and reliability of these parameters may be questionable and highly dependent on the hardware used, and algorithms applied [17]. Thus, the factors mentioned above are not always clearly defined or explained; yet, the parameters are still widely used to quantify the general population’s activity and the external load and internal parameters of team sport athletes.

However, keeping track of loading in team sports is a complex task: Running-based team sports are intermittent sports, consisting of hundreds of brief and very intense actions, such as jumps, tackles, changes of directions, accelerations, and decelerations [18]. These movements are metabolically and physically demanding, more than the same distance covered at a constant speed [19]; thus, specific approaches to quantifying loads for team sport athletes are needed.

Consequently, sports scientists and tracking device manufacturers have created several parameters such as “PlayerLoad™” (PL), “impact load”, or “leg stiffness”, intending to capture load characteristics and their changes with, e.g., fatigue or training status. One of the main challenges in developing load parameters is to capture the demands of accelerating and decelerating, as well as turns and tackles. “Metabolic power”, for example, is one more recently developed approach that attempts to capture the demands of accelerating based on the assumption that this is comparable to the metabolic demands of running uphill [20]. Nevertheless, it does not capture the lateral movements, turns, and tackles.

New possibilities have been created using MEMS to quantify loading in team sports athletes. Nevertheless, a consensus on quantifying the “internal” load of team sport athletes by “external” locomotor measurements is still missing [1,21,22,23]. Consequently, common ground for best practice in load monitoring of team sport athletes has not been established so far [1,22]. In particular, detailed knowledge about the relationship between a recorded external load and internal parameters is rare. A recent meta-analysis has analyzed the relationship between external and internal load parameters in team sport athletes [24]. This work focused on the relationship between HR indices, RPE, and various external load parameters. However, as outlined above, beyond internal load, a multitude of internal processes are stimulated, which are relevant for the psycho-physiological response and adaptation to exercise, as well as the risk of injury.

Consequently, two main challenges regarding load monitoring in team sport athletes have been identified: First, the complex task of quantifying the complex loading situation of intermittent team sports, and second, the difficulty of knowing the relationship between the given MEMS-based external load and the athlete’s individual internal loading and consequently exercise-responses and adaptations within different domains. Identifying these relationships offers great potential to improve the understanding of individual load-response profiles.

Therefore, this systematic review addresses MEMS-based external load parameters and their relationship to various internal parameters, encompassing biochemical, neuromuscular, subjective, cardiovascular, and further domains. This work could aid practitioners in choosing and interpreting appropriate parameters to monitor load in a time- and cost-effective manner to provide the appropriate stimuli to induce adaptations to improve sports performance and decrease the risk of injury.

## 2. Materials and Methods

### 2.1. Article Search, Inclusion, Exclusion

A systematic literature review was conducted based on the preferred reporting items for systematic reviews and meta-analyses (PRISMA) guidelines [25]. The following electronic databases were searched: PubMed, Web of Science, SPORTDiscus, and CINAHL. The search term was created by linking four sections with the Boolean operator “AND”, ensuring that at least one word from each section will appear in the results. Keywords within one section were connected with the operator “OR”. The first section contained various team sports. The second section contained methods and systems used to monitor athletes. The third and fourth sections contained numerous external and internal or performance measurement parameters. Truncation searching was employed to find variations of certain words (see Table 1 for the complete search term). The databases were searched with no restrictions from the earliest records available up to September 1, 2020. Results were stored in a citation manager, and all duplicates were removed (search process see Figure 2). All abstracts were then screened for eligibility regarding the inclusion and exclusion criteria assessed. Any studies including athletes younger than 18 years were excluded as cognitive development influences the accuracy of the RPE [26]. Articles were considered if they showed a relationship measure between one external and one internal or performance parameter obtained from able-bodied team sport athletes during regular training or match play which did not include additional interventions, such as nutritional interventions or manipulated play. For the complete list of inclusion and exclusion criteria, please refer to Table 2. All data were independently extracted by two researchers (JH, JD). In case of disagreement, a consensus was found by a third reviewer (KR). The study further adheres to the ethical standards in sports and exercise science research [27].

### 2.2. Study Quality Assessment

After the final selection was made, the quality of the selected studies was assessed using a 16-item checklist developed by Law et al. [28] and modified by Sarmento et al. [29], which has also been used in previous reviews [29,30]. The authors are aware that a risk of bias assessment may be superior to a checklist summarizing components into a single number, especially when concerned about randomized controlled trials. This systematic review, however, is concerned with observational studies, and thus, the authors decided on a quality checklist that is applicable to the topic at hand. The items on the checklist were scored on a binary scale (0 = no, 1 = yes). Items 6 and 13 included the option “not applicable”. The sum of the scores for each study divided by the maximum value possible for that study represented the quality score. Expressed as a percentage, the score then indicated the methodological quality of the studies. The following meaning was associated with the final percentage: low methodological quality ≤ 50%; good methodological quality 51–75%, excellent methodological quality > 75%. The quality assessment was carried out independently by two reviewers (JH and KR). Disagreement was solved by discussion.

### 2.3. Data Extraction

Data extraction was done using a custom-made sheet pilot tested on five randomly chosen articles. The sheet was redefined, and its final version was used by one reviewer (JH) who performed the data extraction. In case of unclear or missing data, corresponding authors were contacted. The following data were extracted from the studies: (1.) The type of team sport; (2.) the study sample (along with the number of participants, gender, and level/league athletes competed in); (3.) the external parameters recorded or calculated; (4.) the internal parameters measured or calculated and/or the fitness assessment; (5.) the relationship between the external and internal parameters as indicated by statistical association or predictive measures.

### 2.4. Data Synthesis

Data were categorized into groups consisting of the different team sports. Then, subgroups according to the parameters analyzed were created. The subgroups are based on Figure 1 and consist of: the assessment of the relationship between external load parameters and internal load collected during exercise, exercise-induced responses, adaptations, and individual characteristics.

A descriptive synthesis was undertaken with the data structured in a table containing the team sport, the studies included, the load parameters collected, including their frequency of use per sport, and the statistical relational measures between the external and internal parameters. The overall frequency of use of each external and each internal parameter was visualized using pie charts.

## 3. Results

### 3.1. Search Results

The initial search returned 3573 articles. A total of 2234 records remained after removing duplicates; these articles were screened by title and abstracts against the eligibility criteria. After further exclusion of studies (*n* = 2178) that did not meet the criteria, 66 articles remained for the final analysis (Figure 2). The main reasons for exclusion were not using MEMS-based parameters, not analyzing regular training or match play, and analyzing only internal or only external parameters. The references of the included articles were screened, but no further study met the inclusion criteria. The mean methodological quality score of the included studies was 84.6% (+/−8.4%). No article was excluded due to low quality. Ten studies scored between 51 and 75% as good methodological quality. The remainder (*n* = 56) qualified as excellent regarding methodological quality. The most common item to lose quality points on was item 5: justification of the study sample size.

### 3.2. Basic Characteristics of Included Studies

The articles included in this systematic review ranged from 2011 to 2019. The sports analyzed were: American football (*n* = 6), Australian football (*n* = 11), basketball (*n* = 4), field hockey (*n* = 1), rugby union and rugby league (*n* = 8), soccer (*n* = 35), and tag football (*n* = 1). The participants were professional (*n* = 606), elite (*n* = 413), college/university (*n* = 402), and semi-professional (*n* = 120) athletes. *n* = 62 studies included male participants, totaling 1479 male athletes. Four studies studied female participants, accounting for *n* = 62 female athletes. The three most commonly external parameters recorded were distances in speed zones (*n* = 55), total distance (*n* = 46), and PL (*n* = 34), as depicted in Figure 3. The most frequently recorded internal parameters were (session) RPE (this includes RPE as well as session RPE and thus termed “(session-)RPE” going forward) (*n* = 29), HR-based indices (*n* = 19), and well-being questionnaires (*n* = 17), as depicted in Figure 3.

### 3.3. External and Internal Parameters

About 34 external and 32 internal parameters and parameter groups were included across all studies. Different HR-based and various (session-)RPE parameters were grouped and displayed in Figure 3 and Figure 4. The most often investigated external parameter was distance covered in specific speed zones (*n* = 55) which was investigated in 82% of studies included in this review, followed by total distance (*n* = 46), analyzed in 67% of the research articles in this review, and PL (*n* = 34), occurring in 51%. (session-)RPE (*n* = 29) was most often investigated amongst the internal parameters, followed by HR-based indices (*n* = 19) and well-being questionnaires (*n* = 17), occurring in 45, 28, and 27% of research articles included in this systematic review, respectively. From the 66 articles included, 109 different relationships between external and internal parameters have been extracted. The most frequently analyzed relationship was between (session-)RPE and PL with predominantly moderate to strong associations (r = 0.49–0.84). The second most frequently analyzed relationship was between (session-)RPE and distances in speed zones with heterogeneous results. All results for the 109 relationships can be found in Appendix A.

### 3.4. Summary of Individual Studies

Table 3 provides an overview of all studies included in this systematic review, grouped by sport. It includes the number of participants, their playing level, and the collected external and internal parameters. Study designs, participants, hard- and software used, and outcome measures varied noticeably such that the authors focused on describing the results of the studies rather than performing a meta-analysis. Appendix A further shows the relationship measures between parameters.

## 4. Discussion

This systematic review aimed to enhance the knowledge around relationships between external, wearable-based load parameters and internal load, exercise-induced responses, adaptation parameters, and parameters of individual characteristics in running-based team sports. Knowledge about these relationships may reduce time- and possibly cost-intensive testing outside regular training. Acute fitness and fatigue states may be drawn only based on external load parameters. Additionally, the amount of data to be collected and analyzed could be reduced by collecting fewer internal parameters.

Our systematic review is the first to include a myriad of external and internal parameters, focusing on external parameters collected from wearables only. This is crucial to enhance practicality and usability of parameters collected on-field. As the amount of data from wearable sensors and their use increase, it is inevitable to enhance the knowledge around these parameters and understand the dose–response relationship of team sport athletes. The findings are discussed in the following sections.

As some relationships have been examined by a minimal number of studies, results are discussed only when a systematic synthesis of results is feasible. In the following, results are discussed in categories of internal load, exercise-induced response, adaptation parameters, and individual characteristics (Figure 1). Figure 4 additionally highlights the findings of moderate to large relationships which are explored in the following sections, separated by sports. Then, general aspects and future directions are discussed and outlined.

**Figure 4 sensors-23-00827-f004:**
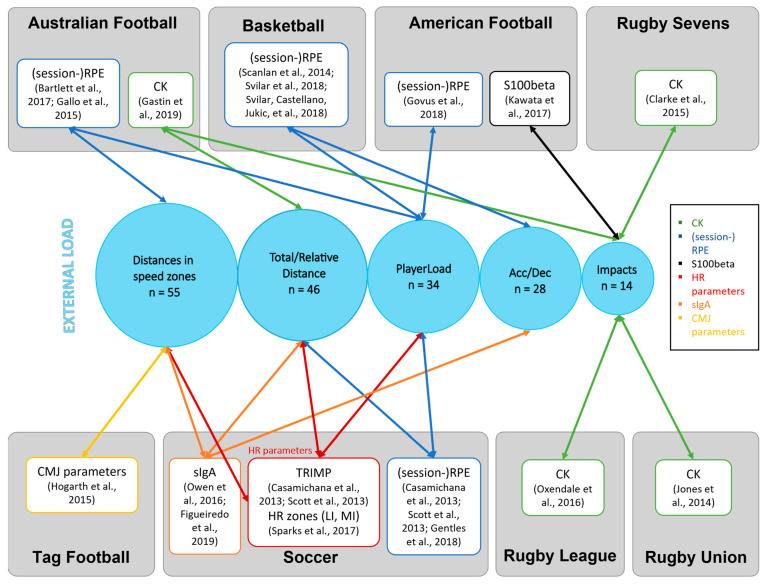
Displayed are parameters for which a systematic synthesis was feasible and that exhibited a moderate to strong relationship. Internal load parameters are sorted by sport and colored as indicated in the legend. The relationship to an external load parameter is marked by an arrow. External parameters are displayed with the number of studies they are appearing in [13,31,32,38,39,43,47,48,49,51,56,57,59,60,69,80,91,93,94]. Acc/Dec acceleration/deceleration parameters, CK creatine kinase, CMJ countermovement jump, HR heart rate, RPE rating of perceived exertion, sIgA secretory immunoglobulin A, TRIMP training impulse, LI low intensity, MI medium intensity.

### 4.1. Internal Load

#### 4.1.1. (session-)RPE

Internal load parameters predominantly encompassed subjective ratings of exertion and HR-based indices.

(session-)RPE had moderate to strong associations with total and relative distance in Australian football [13,43] and soccer players [59,60,93]. Moderate to strong associations were also present between (session-)RPE and PL in Australian football [43], basketball [47,48], and soccer [59,60]. Similar strength of associations was detected between (session-)RPE and distances covered in different speed zones in Australian football players [43], and acceleration and deceleration parameters in basketball players [48,49]. A weak relationship between (session-)RPE and PL was only found in American football players [31], and heterogenous results were present for the relationship between (session-)RPE and distances covered in speed zones and (session-)RPE and acceleration and deceleration parameters in soccer players.

The weak relationship between (session-)RPE and PL in American football may be due to different match demands compared to other team sports analyzed. American football players generally cover lower overall distances in games (3000 to 5500 m in NCAA I football [59]) than soccer players (male elite outfield players 9000–14,000 m [95]), Australian football players (elite level 12,939 ± 1145 m [96]), and even basketball players (4404 to 7558 m [97]). Bartlett et al. (2017) found collinearity between session duration, PL, and total distance. This may suggest that the lower the overall distance, the lower the association between PL and RPE or session-RPE.

Heterogenous results between (session-)RPE and distances in speed zones for soccer players may be due to the larger volume of studies compared to other team sports included and due to the different methods used: partial correlations [92], within-individual correlations [73,92], Pearson product-moment correlations [59,60], and machine learning techniques [61,64]. Furthermore, speed thresholds to define speed zones were either fixed or individualized, and distances were expressed in various ways (as absolute, percentage of total distance, frequency, number of efforts, or distance per minute).

Generally, indicators of total volume seem to result in higher associations than those expressed per minute or as a percentage of total volume. Rago et al. (2019) found a tendency of increasing correlations when speed thresholds were individualized rather than identical for all players.

Heterogenous results between (session-)RPE and parameters describing accelerations and decelerations in soccer may be due to varying methods. Using partial and within-individual correlations [92], small to moderate correlations were detected between those parameters. Furthermore, correlations describing total acceleration were higher than those describing accelerations per minute [92]. This supports the above findings that (session-)RPE seems to have stronger associations with parameters describing total volume. Machine learning techniques identified the number of acceleration efforts and decelerating distances as the main contributors to RPE in soccer [64].

Correlations between (session-)RPE and acceleration and deceleration parameters may be higher in basketball due to the high frequency of accelerations of 29.6 ± 3.9–32.7 ± 11.0 per minute in professional male players [98] compared to 90 ± 21 total accelerations per match in soccer players [99]. This places a greater total amount of accelerations and decelerations on basketball players compared to i.e., soccer. Thus, accelerations and deceleration may have a greater impact on perceived exertion compared to team sports in which they occur less frequently.

For practitioners, this means that estimation of (session-)RPE may be done most adequately with indicators of total volume such as total distance or PL in Australian football, soccer, and basketball players. In indoor team sports such as basketball, where total distance is not available from wearable sensors due to a lack of GNSS signal, parameters describing acceleration and deceleration may be used instead of total distance. Omitting (session-)RPE scales would save practitioners and players the time to analyze and fill out the scales.

#### 4.1.2. HR-Based Indices

The relationships of HR-based parameters of internal load to external load parameters were analyzed in soccer players only. HR was divided in zones [79,80], predicted from external load parameters [65], expressed as percentage of HR_max_ [79,84], and used for calculations of Edward’s and Banister’s TRIMP [59,60,72,79]. TRIMP is a method, originally proposed by Banister et al. [100,101] that integrates training duration, maximal, resting, and average exercise HR, and a weighting factor to address high intensities [102]. Banister’s TRMIP has further been modified, including a summated HR zone method proposed by Edwards [103], here Edwards’ TRIMP. This method takes the time spent in predefined HR zones into account. The result of both methods is a training score per session indicating the cardio-vascular demands experienced by the athlete. Moderate to large correlations existed between time spent in the low- and medium-intensity velocity and the low- and medium-intensity HR zones, respectively. Time spent in the low- and medium- intensity HR zones also showed moderate to large correlations with PL [80]. Similar strength in associations was found between total distance and Edwards’ TRIMP [60] and between PL and Edwards’ and Banister’s TRIMP [59,60]. Correlations were not significant or weak between time spent in the high-intensity velocity and the high-intensity HR zone [80], between high-speed distance and number of efforts at sprinting speed and Edwards’ TRIMP [60], between PL and the high-intensity HR zone [80], between the number of rapid accelerations per minute and the time spent above 80% of HR_max_ [79], and between repeated high-intensity events and Banister’s TRIMP [79] and percent time spent above 80% of HR_max_ [79]. With increasing speed, correlations between HR-based parameters and distances in speed zones seem to weaken. This finding is similar to (session-)RPE which may be due to the high relationship between HR-based parameters and session-RPE [104,105]. Previous research has highlighted that HR measures may not be appropriate for high-intensity interval training or intermittent exercise due to the increase in anaerobic contribution [102,106]. For practitioners, this means that high-intensity running parameters may not be an appropriate indicator of HR. Low-intensity running and indicators of total volume such as total distance or PL may be more suited for low- and medium-intensity HR parameter estimation.

Overall, among the internal load parameters, (session-)RPE, time spent in low- and medium-intensity HR zones, and TRIMP are best estimated using parameters of total volume such as PL and total distance. Time spent in high-intensity HR zones are not represented adequately by the external load parameters examined and may be recorded separately if of interest. Noteworthy is that HR-based indices do not adequately represent anaerobic training [102,106]. The latter two points support the idea of collecting both HR-based indices and external load parameters.

### 4.2. Exercise-Induced Responses

Exercise-induced response are short-term changes in parameters. To detect changes, parameters are collected at two or more time points several hours apart. The first time point serves as a baseline measure, usually in a non-fatigued state. The next time point(s) occur(s) following exercise when athletes may be fatigued. Exercise-induced responses extracted from the studies included consist of well-being indicators, CK concentrations, HR-based indices, neuromuscular functioning, and biomarkers, among others.

The relationships between well-being parameters and external load indicators were heterogeneous, possibly due to studies using different questionnaires analyzing either overall wellness [35,36,38,50,68,76] or single parameters of wellness (e.g., sleep, stress, recovery, muscles soreness, or fatigue) [56,83,84,87,88] and related it to external load parameters of the same day [38,50,56,76], the previous day [35,36,63,83], the previous two to four days [36,84], or the previous weeks [88]. Thus, practitioners may need to be careful when choosing the specific parameter and the time period being analyzed as this provides different information about the athlete.

Results in Appendix A indicate that especially high-intensity distance parameters such as distance covered, time spent, or number of efforts at high speeds may inform about potential muscle damage as indicated by CK levels. In collision team sports such as Australian football, rugby league, rugby union, and rugby sevens, moderate to strong relationships were found between CK levels and impact parameters [39,51,56,57]. In these sports, impact parameters may additionally indicate muscle damage. No study included addressed those parameters in American football players. However, since American football belongs to the collision team sports, similar associations to CK levels can be expected. This, however, needs further verification.

Exercise-induced HR-based indices were found only in studies observing soccer players. Here, mostly negligible to small associations between several external load indicators and heart rate variability (HRV) were found [71,83,84]. For practitioners, this means that common external load indicators, as included in this review, may not serve as an estimation of HRV. Thus, this metric should be recorded separately if it is of interest.

The relationship between change in countermovement jump (CMJ) parameters as an indicator of neuromuscular fatigue and high-speed running parameters varied from negligible and nonsignificant to large and was assessed in soccer and tag football players. Three studies found negligible to small correlations [83,84,86], whereas another three studies found moderate to large correlations [74,75,94]. This may be due to the different CMJ parameters collected (jump height, GRFs, or power output) and different time points when fatigued jumps were executed (24 to 72 h post-exercise).

CMJ parameters’ relationship to acceleration and deceleration parameters exhibited varying results. One study assessed relationships to parameters of overall volume, such as total distance, duration, and PL, only finding trivial or unclear effects in soccer players [86]. For practitioners, this means that a high volume of acceleration, deceleration, and high-speed running efforts possibly negatively influence neuromuscular performances for up to 24 h. Further research is needed to assess CMJ parameters relationships.

In American football, the number of impacts and peak head accelerations may indicate S100beta levels but not tau concentrations [32]. Here, data were collected from an instrumented mouthguard. S100beta is a blood biomarker that may be useful in detecting mechanical stress in the brain [32]. In the field, symptom scores offer a quick and easy-to-use method to detect symptoms of concussions [107]. Kawata et al. (2017) did not find higher symptom scores in players who sustained more impacts. This finding is particularly important to recognize for practitioners as the easy-to-use method (symptom scores) may fail to detect exposure to repeated sub-concussive (head) impacts. Accelerometer-embedded mouth guards, however, seem to pose a reasonable method to detect elevated S100beta levels. Thus, it may be beneficial to implement external monitoring systems such as accelerometers. Further studies are needed to analyze these relationships.

Concentrations of secretory immunoglobulin A (sIgA), a marker of immune function, have been linked to high-intensity distance, total distance, and acceleration and deceleration parameters in soccer [69,91]. Small to large negative relationships were observed. If the overall volume is high, sIgA is reduced; thus, immune function and the risk of contracting an upper respiratory tract infection (URTI) may be increased. This finding is in line with the previous findings that reported reduced sIgA levels after interval runs [108] and in athletes with a high workload [109]. For practitioners, this means that players are at a higher risk of falling sick following a period of congested schedule or high volume in general. Players may be at greatest risk 3 to 72 h post intense exercise according to the “open window theory” of altered immunity [109].

Overall, numerous different exercise-induced responses were analyzed, as previously depicted in Figure 3. Most of them, however, appeared in few studies such that a systematic synthesis was not feasible. For the parameters discussed above, practitioners need to carefully consider time point of collection and the specific parameter collected, as changes may result in varying outcomes. Consistent findings were present regarding muscle damage in collision-based team sports which may be estimated using impact parameters. Practitioners shall further consider that sIgA levels may be low following high total activity volume and players may be at risk of attracting an URTI.

### 4.3. Adaptation Parameters

Adaptation parameters are assessed at a minimum of two time points several days, weeks, or months apart and they are commonly carried out in a non-fatigued state. Changes between measurements can be analyzed and viewed as adaptation. Adaptation parameters extracted from the studies included were related to changes in body mass, HR-based indices, intermittent and aerobic endurance capacities, sleep efficiency, and strength parameters.

Body mass seems to change in relation to 10-week sprinting distance, and total distance, but not session duration or average speed [78,82]. Change in HR_max_ was positively related to 10-week sprinting distance [82], and HRV negatively to acute-to-chronic workload ratio (ACWR)-based session time during a season [77]. ACWR is commonly used for injury prevention purposes. The model says that a greatly increased or decreased acute workload, compared to the chronic workload, increases the risk of sustaining an injury [110]. Change in hamstring peak torque, quadriceps to hamstring ratio, percent change in peak torque, and quadriceps to hamstring ratio was positively related to 10-week accumulated PL and acceleration sum (accumulated acceleration data in all three axes), sprinting distance, duration, and total distance, respectively [82]. Meaning volume and intensity improve strength test performances and may thus reduce hamstring injuries and the risk of suffering an anterior cruciate ligament (ACL) injury as imbalances between quadriceps and hamstring constitute an ACL risk factor [111]. Monitoring loads long term to ensure they are sufficiently high to cause adaptations may be beneficial. Further, team sports practitioners can gain a better understanding of the individual dose-response patterns.

As indicated by a positive change in VO_2max_, improvements in aerobic endurance were strongly correlated to 10-week accumulated PL and acceleration [82]. Session duration, however, showed an inverse relationship to changes in VO_2max_ [82]. The duration in this study, however, decreased throughout the season. Meaning VO_2max_ increased despite decreasing duration. In this case, other factors, such as high mechanical loading, seem to represent improvements in aerobic capacity better than training duration.

Changes in intermittent fitness, as indicated by the 30-15 intermittent fitness test, were observed by one study only, which found unclear and large relationships to high-intensity running, total distance, and PL, respectively [72]. More research is needed regarding these relationships to synthesize results systematically.

Overall, findings suggest that intensity seems particularly important to improve certain physiological capacities related to intermittent team sports. Volume and intensity need to be well-balanced in training programs to cause optimal adaptations.

### 4.4. Individual Characteristics

Individual characteristics analyzed in relation to external parameters collected using wearables include intermittent and aerobic endurance capacity, neuromuscular performance parameters, and muscle architecture. As indicated by the Yo-Yo intermittent recovery test (YYIR), players with larger intermittent endurance capacities covered greater total distances in soccer and tag football [66,94]. Greater intensities, as indicated by high-speed running meters per minute and the number of repeated high-intensity events, were covered by tag football players with better YYIR performance [94]. Similar findings were present for players with greater VO_2max_ regarding total distance and intensity parameters in soccer players [67]. For practitioners, this means that players who generally cover more distances likely have greater endurance capacities. As such, rigorous and time-intensive testing in the laboratory may not be necessary to find out endurance deficits and strengths in players.

Parameters of neuromuscular performance and muscle architecture were analyzed each by one study only; thus, a systematic synthesis is not feasible, and more research is needed to draw conclusions regarding those parameter relationships.

### 4.5. General Aspects

Generally, based on the intensity and volume of external load experienced during training and match play, internal bodily reactions take place, which, in the long term, lead to adaptations and influence individual characteristics. Those characteristics determine how well a player can handle the external load. However, no consensus exists on the parameters encompassing internal load yet. Some researchers have a broad understanding of internal load parameters, including biochemical, neuromuscular, and hormonal responses [1,2]. Others have a more narrow definition of internal load, including only measures that can prescribe exercise intensity, comprising mainly HR and session-RPE [3,4,110]. In this systematic review, parameters were categorized mainly based on the time point of measurement, as depicted in Figure 1. This, however, was not always straightforward: Couderc et al. (2017) collected blood from the fingertip 3 min post-exercise to determine lactate and bicarbonate concentration and pH levels. As those parameters are not monitored continuously during exercise, they fall into the category of exercise-induced responses. RPE, however, is commonly collected around 15–30 min after exercise [53,59,73,92] to reduce bias that may result from particular easy or challenging segments during the final exercise period [112]. However, RPE is deemed an internal load parameter by both parties of broad and narrow definitions of internal load. This might be due to the strong relationship of RPE with internal load parameters (HR and blood lactate) [113,114].

Some limitations to this systematic review are acknowledged in the following. These include the non-feasibility of synthesizing results for some parameter relationships. Given the wide variety of parameters, some relationships were analyzed by fewer studies to synthesize results systematically. Further, thresholds for speed zones differ across studies such that results may vary due to varying absolute or individualized thresholds used. Different hard- and software was implemented in the studies analyzed, which may cause a discrepancy in results. Manufacturers apply filters to the data during post-processing such that the same parameter could supposedly differ when obtained from another product. Even a software update could result in inconsistent results.

Correlations found do not mean causality; parameters might correlate because of other circumstances. Clemente et al. (2019) found a large negative correlation between training duration and VO_2max_. This finding, however, likely does not mean shorter training durations cause an increase in aerobic endurance, but rather other circumstances were in place, such as high running volume and repeated high-intensity events, that may elicit improved aerobic endurance.

Few studies (*n* = 4) included in this systematic review studied female athletes [46,51,67,93], totaling 62 female athletes compared to 1479 male athletes. Even though the parameter relationships were comparable between males and females, this can only be said about the few parameters’ relationships analyzed. Thus, more research, including female athletes, is needed.

### 4.6. Outlook and Future Work

Besides ever ongoing enhancements in hard- and software that will provide more accurate and reliable data in the future, research around accelerometer- or inertial-based GRF and moment estimation has shown promising results. Estimating GRFs and joint moments during training and match play would provide valuable biomechanical insight. Continuous monitoring of forces and moments acting on the player’s body could enhance the knowledge of the optimal individual dose-response relationship, injury mechanisms, and performance indicators from a biomechanical perspective. Research groups have placed MEMS on the shank [115], the sacrum [116], and the trunk [117] or used a full-body sensor suit [118]. So far, movements such as walking, jumping, running, and squatting, have been analyzed separately. This approach needs further development for more complex and compound movements to be transferrable to team sports. Players will either have to wear sensors in more locations (possibly embedded in clothing), or algorithms based on one trunk-mounted sensor only will have to be developed further to gain valuable insights into the said domains.

Recently, advancements in continuous lactate and glucose monitoring have been made. This methodology will provide new insights into the external–internal load relationships of those parameters which need to be studied in the future. More possibilities to monitor internal load parameters continuously during training and match play as well as knowledge about the relationships to external parameters will move testing and identification of adaptations and health and fitness status away from the laboratory and more toward on-field assessments. Thus, separate time-consuming and fatiguing testing can be eliminated and replaced with data collected during training and match play.

An ever-increasing amount of data will be collected, so the ability to analyze and select data appropriately according to the context becomes increasingly important. Even though some external parameters show strong relationships with, e.g., HR, it is essential to use parameters according to their context. If a highly anaerobic training session takes place, valid measures may differ from those of a more aerobic-based session. Despite strong correlations between parameters, and even if both external and internal parameters are considered, it is still essential to know what type of parameters to inspect depending on the demands placed on the athletes and the stressed biological systems. Having a sound understanding of those differences is inevitable for practitioners to harness the power of the collected data. Besides selecting parameters, verbal and visual transfer of information becomes increasingly relevant to create a common understanding between athletes, coaches, data analysts, and medical staff to enhance performance.

Future work will need to validate novel methods of collecting internal parameters, analyze relationships of those to external parameters, and include more female athletes. With more reliable data, captured from highly dynamic movements, the impact of those movements on players can be explored in more depth, as the parameter relationships in this domain are currently ambiguous.

## 5. Conclusions

Strong correlations have been detected, especially between parameters of total activity volume and the internal load parameters HR-indices and RPE or session-RPE. These parameter relationships, were analyzed most often making the state of evidence clearer than less researched parameter relationships. Fitness tests assessing aerobic and intermittent endurance, or (session-) RPE, and in collision-based team sports, additionally markers of muscle damage may be omitted and replaced by external, on-field measurements, facilitating the work of practitioners. Relationships between external load and the other three internal parameter categories, exercise-induced responses, adaptations, and individual characteristics, are mostly ambiguous and need further verification. Until then, a holistic picture of an athlete may best be obtained by collecting external and internal parameters for those parameter groups. Due to the ever-increasing amount of data collected in both areas, external and internal, a sound understanding of the data and their sport-specific context becomes increasingly important. Good communication is crucial for all stakeholders to attain a common understanding of the data. Studies including female athletes have been noticeably little in number and should be increased in the future. Future work will need to validate novel methods of collecting internal parameters and their relationships to external parameters in order to understand the individual dose-response patterns.

## Figures and Tables

**Figure 1 sensors-23-00827-f001:**
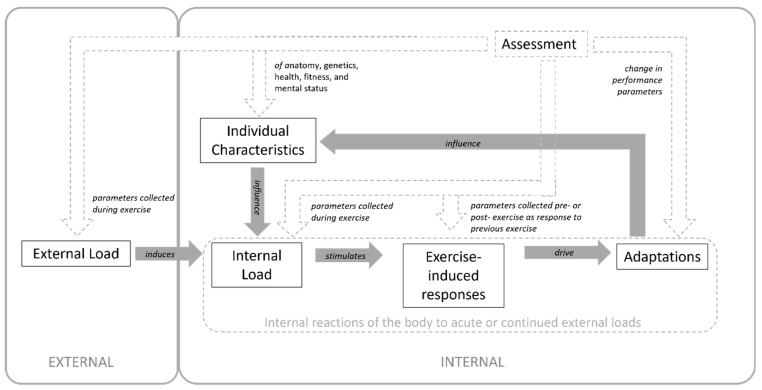
Interaction of external and internal parameters and possibilities to assess those parameters.

**Figure 2 sensors-23-00827-f002:**
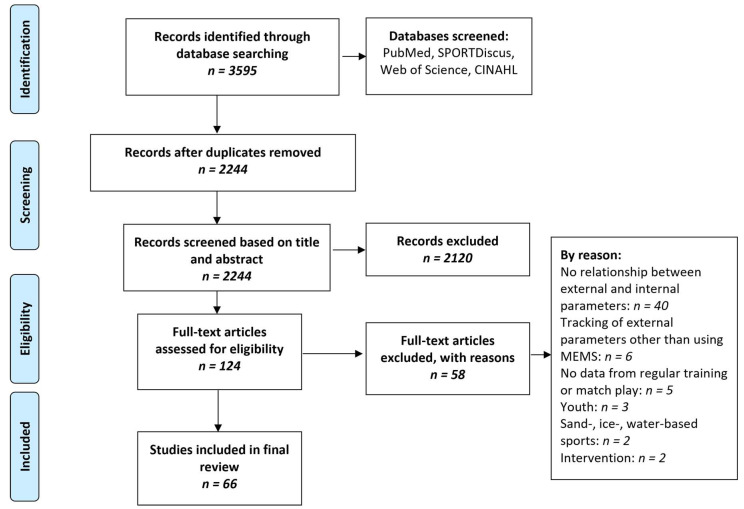
Flow diagram of the study selection process adopted from the PRISMA guidelines.

**Figure 3 sensors-23-00827-f003:**
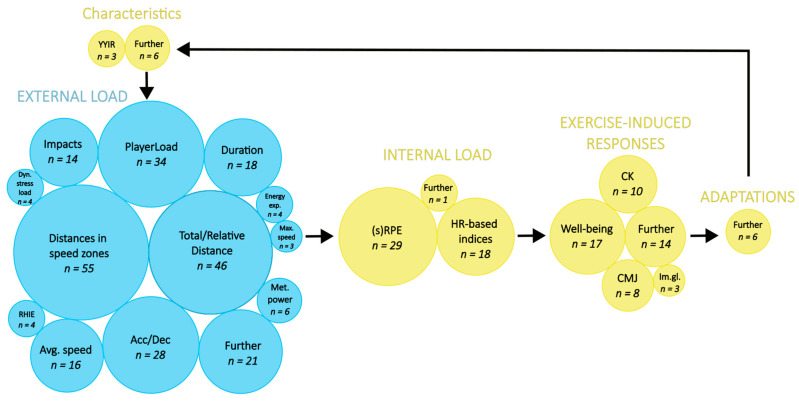
External and internal parameters with the number of studies they are appearing in. Parameters occurring in one or two studies only are pooled under “Further”. Acc/Dec, acceleration and deceleration parameters; Avg., average; CK, creatine kinase; CMJ, countermovement jump; Dyn., dynamic; exp., expenditure; HR, heart rate; Im.gl., immunoglobulin parameters; Max., maximal; Met., metabolic; RHIE, repeated high-intensity efforts; (s)RPE (session) rating of perceived exertion; Well-being, well-being questionnaires; YYIR, Yo-Yo intermittent recovery test.

**Table 1 sensors-23-00827-t001:** Search Term: Categories are connected with the Boolean operator “AND”; key words within a category are connected with “OR”.

Category	Keywords
Team Sport	“Team Sport*” OR soccer OR football OR handball OR basketball OR rugby OR volleyball OR futsal OR netball
Monitoring system	monitoring OR tracking OR GPS OR “Global Positioning System”[MeSH] OR LPS OR “Local Positioning System”[MeSH] OR IMU OR “inertial measurement unit” OR acceleromet* OR MEMS OR microsensor OR “time motion” OR TMA OR “motion analysis”[MeSH] OR “wearable technologies”[MeSH]
External load	workload OR load OR speed OR ACWR OR “acute to chronic work ratio” OR “work:rest” OR distance OR acceleration OR “metabolic power” OR “metabolic load” OR PlayerLoad OR intensit* OR “energy expenditure” OR “high intensity burst*” OR “work ratio” OR “fatigue index” OR “physical” OR “repeated sprintability
Internal load	“internal load” OR RPE OR “rate of perceived exertion” OR RPE OR sRPE OR “heart rate” OR HR OR TRIMP OR questionnaire OR biochemical OR physiological OR neurological OR fatigue OR blood OR lactate OR SPX OR Spiroergometry OR “breath gas analysis” OR CK OR “creatine kinase” OR VO2 OR “anaerobic threshold”

**Table 2 sensors-23-00827-t002:** Inclusion and exclusion criteria.

Inclusion	Exclusion
Topic of the article is human physical performance	Topic not related to physical performance or non-human subjects
Original research	Surveys, opinions, books, case studies, non-academic text, reviews, conference abstracts
Competitive field- or court-based team sport athletes	Individual sports, ice-, sand-, or water-based team sports, referees
Adult athletes	Athletes under 18 years of age
Able-bodied, non-injured athletes	Special populations (i.e., clinical), mentally or physically impaired athletes, injured athletes
Training or match play	Laboratory settings, and field-based settings coupled with an intervention (i.e., nutritional intervention).
Report of at least one external and one internal load measure or physiological fitness assessment	Report of only internal or only external measures
Report of a relationship between internal and external measures	No relationship between internal and external measures reported
Use of GNSS, MEMS, IMU, LPS	Use of timing gates, measuring tapes, video-based tracking
Good, very good, or excellent methodological quality based on the checklist used for this review	Poor methodological quality based on the checklist used for this review

**Table 3 sensors-23-00827-t003:** Studies included in this systematic review sorted by type of team sport. The table includes information about the player level and the parameters collected.

Sport	Study	Player Level (*n* = Number of Athletes)	External Parameters (*n* = Number of Studies)	Internal Parameters (*n* = Number of Studies)
American football	[31,32,33,34,35,36]	University Divison I (*n* = 225, male)	PL (AU) (*n* = 4) Acceleration/Deceleration (m·s^−2^) (*n* = 4) Distance in speed zones (m) (*n* = 2) Impacts (*n*) (*n* = 2) Stride variability (*n* = 1)	INTERNAL LOAD PARAMETERS (session-)RPE (AU) (*n* = 1) EXERCISE-INDUCED RESPONSES Well-being questionnaire (5-point scale) (*n* = 4) S100beta (pg/mL) (*n* = 1) Tau concentration (pg/mL) (*n* = 1)
Australian football	[13,37,38,39,40,41,42,43,44,45]	Professional (*n* = 202, male) Elite (*n* = 118, male)	Distance in speed zones (m) (*n* = 13) PL (au) (*n* = 9) Total/Relative distance (m, m/min) (*n* = 9) Duration (min) (*n* = 5) Average speed (m/s) (*n* = 4) Acceleration/Deceleration (m·s^−2^) (*n* = 3) Energy expenditure (kJ/kg) (*n* = 2) Metabolic power concept (W/kg) (*n* = 2) Distance load (m^2^/s) (distance x mean speed) (*n* = 1) Effort zones (*n*) (*n* = 1) Equivalent distance (m) (*n* = 1) Explosive efforts (*n*) (*n* = 1) Impacts (*n*) (*n* = 1) Match exercise intensity (AU) (*n* = 1)	INTERNAL LOAD PARAMETERS (session-)RPE (AU) (*n* = 7) Core temperature (C) (*n* = 1) EXERCISE-INDUCED RESPONSES Well-being questionnaire (5-point scale) (*n* = 3) CMJ (cm) (*n* = 1) CK (U/L) (*n* = 1) INDIVIDUAL CHARACTERISTICS Maximal aerobic speed (m/s) (*n* = 1) YYIR (m) (*n* = 1)
Basketball	[46,47,48,49]	Elite (*n* = 12, male) Professional (*n* = 26, male) Semiprofessional (*n* = 8, male) University (*n* = 5, female)	PL (AU) (*n* = 4) Acceleration/Deceleration (m·s^−2^) (*n* = 4) Jumps (*n*) (*n* = 2) IMA™ (AU) (*n* = 1)	INTERNAL LOAD PARAMETERS (session-)RPE (AU) (*n* = 3) HR-based indices (*n* = 1) EXERCISE-INDUCED RESPONSES Tensiomyography (ms, mm) (*n* = 1)
Field Hockey	[50]	Elite (*n* = 12, male)	Acceleration/Deceleration (m·s^−2^) (*n* = 1) Distances in speed zones (m) (*n* = 1) Total/relative distance (m, m/min) (*n* = 1)	EXERCISE-INDUCED RESPONSES Well-being questionnaire (5-point scale) (*n* = 1)
Rugby Sevens	[51,52]	Elite (*n* = 24, 12 female, 12 male) Amateur (*n* = 10, female)	Total/relative distance (m, m/min) (*n* = 2) Distance in speed zones (m) (*n* = 2) Impacts (*n*) (*n* = 1)	EXERCISE-INDUCED RESPONSES CK (U/L) (*n* =1) Bicarbonate concentration (mmol/L) (*n* = 1) Lactate concentration (mmol/L) (*n* = 1) pH (*n* = 1)
Rugby League	[53,54,55,56]	Professional (*n* = 46, male) Elite (*n* = 45, male)	Distance in speed zones (m) (*n* = 3) Impacts (*n*) (*n* = 3) Acceleration/Deceleration (m·s^−2^) (*n* = 2) Total/Relative distance (m, m/min) (*n* = 2) Duration (min) (*n* = 1) PL (AU) (*n* = 1) RHIE (*n*) (*n* = 1)	INTERNAL LOAD PARAMETERS (session-)RPE (AU) (*n* = 2) EXERCISE-INDUCED RESPONSES Well-being questionnaire (5-point scale) (*n* = 1) CK (U/L) (*n* = 2) Salivary cortisol (nmol/L) (*n* = 1) Repeated plyometric push-ups (*n*) (*n* = 1) Sleep (h) (*n* = 1) ADAPTATION PARAMETERS Sleep (h) (*n* = 1)
Rugby Union	[57,58]	Professional (*n* = 51, male)	Distance in speed zones (m) (*n* = 2) Impacts (*n*) (*n* = 2) PL (au) (*n* = 1) Total/Relative distance (m, m/min) (*n* = 1)	EXERCISE-INDUCED RESPONSES CK (U/L) (*n* = 1) Urinary n-terminal prohormone of brain natriuretic peptide (pg/mL) (*n* = 1)
Soccer	[59,60,61,62,63,64,65,66,67,68,69,70,71,72,73,74,75,76,77,78,79,80,81,82,83,84,85,86,87,88,89,90,91,92,93]	Professional (*n* = 311, male) Elite (*n* = 236, male) Semi-professional (*n* = 61, male) University (*n* = 114, 79 male, 35 female)	Distance in speed zones (m) (*n* = 31) Total/Relative distance (m, m/min) (*n* = 30) PL (AU) (*n* = 15) Acceleration/Deceleration (m·s^−2^) (*n* = 13) Duration (min) (*n* = 12) Impacts (*n*) (*n* = 5) Average Speed (m/s) (*n* = 4) Dynamic stress load (AU) (*n* = 4) Metabolic power concept (W/kg) (*n* = 4) Maximal velocity (m/s) (*n* = 3) Effindex (AU) (*n* = 2) RHIE (*n*) (*n* = 2) Body load (AU) (*n* = 1) Energy expenditure (kJ/kg) (*n* = 2) Equivalent distance (m) (*n* = 1) Explosive distance (m) (*n* = 1) Impulse Load (Ns) (*n* = 1) Force load (AU) (*n* = 1) Mechanical work (AU) (*n* = 1) Training load score by Polar (AU) (*n* = 1) Total accelerometer load (AU) (*n* = 1) Total forces (AU) (*n* = 1) Velocity load (AU) (*n* = 1) Work:rest ratio (*n* = 1)	INTERNAL LOAD PARAMETERS HR-based indices (*n* = 17) (session-)RPE (AU) (*n* = 16) Effindex (AU) (*n* = 2) EXERCISE-INDUCED RESPONSES Well-being questionnaire (5-point scale) (*n* = 8) CMJ (cm) (*n* = 6) CK (U/L) (*n* = 5) Immunoglobulin (μg/mL) (*n* = 3) C-reactive protein (mg/L) (*n* = 1) HR-based indices (*n* = 1) Myoglobin concentration (ng/mL) (*n* = 1) Plasma lactate dehydrogenase (U/L) (*n* = 1) Body mass measures (kg) (*n* = 1) ADAPTATION PARAMETERS HR-based indices (*n* = 2) Body mass measures (kg) (*n* = 2) Strength test (Nm) (*n* = 1) VO_2max_ (ml/kg/min) (*n* = 1) 30-15 intermittent fitness test (m) (*n* = 1) INDIVIDUAL CHARACTERISTICS VO_2max_ (ml/kg/min) (*n* = 1) YYIR (m) (*n* = 1) Repeated sprint ability (m) (*n* = 1) Body mass measures (kg) (*n* = 1) Muscle characteristics (cm) (*n* = 1) Sprint test (s) (*n* = 1)
Tag football	[94]	Regional (*n* = 16, male)	Acceleration/Deceleration (m·s^−2^) (*n* = 1) Distance in speed zones (m) (*n* = 1) RHIE (*n*) (*n* = 1) Total/relative distance (m, m/min) (*n* = 1)	INDIVIDUAL CHARACTERISTICS CMJ (cm) (*n* = 1) Sprint test (m/s) (*n* = 1) YYIR (m) (*n* = 1)

AU arbitrary unit, CK creatine kinase, CMJ countermovement jump, HR_max_ maximal heart rate, HR heart rate, IMA™ inertial movement analysis, PL player load, RHIE repeated high-intensity events, RPE rating of perceived exertion, TRIMP training impulse, VO_2max_ maximal oxygen uptake, YYIR Yo-Yo intermittent recovery test.

## Data Availability

Not applicable.

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
