# Peer review of "Relationships between External, Wearable Sensor-Based, and Internal Parameters: A Systematic Review"

_sensors, 2023, doi:10.3390/s23020827_

Round 1

Reviewer 1 Report

This paper has reviewed the relationship between external sensing parameters and internal load in the application of sport athletes. Adequate references have been searched, referred, and summarized. This is a helpful research for the study of sports science and athletes training. The outline of paper is clear, the conclusions very well addressed the main question, and also propose some challenges or future work in this field. However, I still have a few comments as below to improve the paper:

  1. I would suggest the authors to emphasize more on the Discussion session. The outline is quite clear, but this session lacks a nice figure which can visualize the relationship (and its strength) between the parameters (four categories as stated) and different sports, and make it a network structure. This is to make the part clearer to readers, and conclusions more evident.
  2. I suggest the authors to state the novelty of this review. Except the conclusions made from the references, no new ideas show clearly in the manuscript. Can the authors address this and make corresponding modification in the paper?

Reviewer 2 Report

In this article, the authors have conducted a systematic review of the Relationships between External, Wearable Sensor-Based, and Internal Parameters: a Systematic Review. However, I will comment on some aspects to improve the quality of the manuscript, and the authors must present the highlighted changes:

-The acronyms are misspelt. The correct thing is to write them with capital letters the meaning, such as Heart Rate (HR). This error must be corrected in all acronyms used in this document.

-The parameters collected in the literature do not contain their respective unit, such as "Duration" is it in seconds, minutes or hours? It is necessary to include which unit of measure has been taken into account.

- When referring to an object such as a Figure, Table, Algorithm, Equation, or Section in the text of the document, it must be written completely and with its initial capital letter.

-There is an unnecessary blank space on page 19.

-Conclusions must be improved and future work added.

Round 2

Reviewer 1 Report

The current version is more acceptable. However, I still suggest the authors to consider the network structure to highlight some significant relationships (or internal/external parameters) found in different sports so that readers can quickly grasp the major conclusions from this paper.

Reviewer 2 Report

This article is not in a correct presentation format for reviewers because it has a grey Microsoft Word correction box, and it has even strikesthrough certain content of the manuscript. The authors must correctly and formally present the manuscript, highlighting the changed parts.

Round 3

Reviewer 2 Report

Many thanks to the authors for performing some changes suggested by the reviewers. However, the changes that have not been performed are:

-Misspelled all the acronyms.

-In this manuscript there are two Future Works. The Future Work paragraph must be written at the end of the conclusions and not before.

Without performing these changes by the authors, this article cannot be published.
